# Role of opioid signaling in kidney damage during the development of salt-induced hypertension

Daria Golosova[1,*], Oleg Palygin[1,2,*] , Ruslan Bohovyk[1], Christine A Klemens[1], Vladislav Levchenko[1], Denisha R Spires[1], Elena Isaeva[1] , Ashraf El-Meanawy[2,3], Alexander Staruschenko[1,2,4]

**Opioid use is associated with predictors of poor cardiorenal outcomes. However, little is known about the direct impact of opioids on podocytes and renal function, especially in the context of hypertension and CKD. We hypothesize that stimulation of opioid receptors (ORs) contributes to dysregulation of intracellular calcium ([Ca$^{2+}$]$_i$) homeostasis in podocytes, thus aggravating the development of renal damage in hypertensive conditions. Herein, freshly isolated glomeruli from Dahl salt-sensitive (SS) rats and human kidneys, as well as immortalized human podocytes, were used to elucidate the contribution of specific ORs to calcium influx. Stimulation of κ-ORs, but not μ-ORs or δ-ORs, evoked a [Ca$^{2+}$]$_i$ transient in podocytes, potentially through the activation of TRPC6 channels. κ-OR agonist BRL52537 was used to assess the long-term effect in SS rats fed a high-salt diet. Hypertensive rats chronically treated with BRL52537 exhibited [Ca$^{2+}$]$_i$ overload in podocytes, nephrinuria, albuminuria, changes in electrolyte balance, and augmented blood pressure. These data demonstrate that the κ-OR/TRPC6 signaling directly influences podocyte calcium handling, provoking the development of kidney injury in the opioid-treated hypertensive cohort.**

## Introduction

Excessive opioid use has become a global health concern, yet very little is known about how it affects renal function in chronic kidney disease (CKD) or hypertension. Despite this, almost half of the opioid-prescribed patient population is hypertensive (Herzig et al, 2014). The extensive use of opioid-based pain management strongly correlates with poor cardiovascular and cardiorenal outcomes, including increased albuminuria and reduced glomerular filtration rate (Agodoa et al, 2008; Mallappallil et al, 2017); as such, opioid medication is strongly linked to progressive renal damage and blood pressure elevation (Herzig et al, 2014; Barbosa-Leiker et al,

2016). Despite the growing attention to the opioid problem, our current knowledge of the mechanisms underlying opioid-induced kidney damage is limited and requires detailed investigation to understand better and hopefully prevent associated cardiorenal outcomes. The correlation between the response to repeated opioid exposure and the accelerated progression of cardiovascular diseases has been related to increased prevalence of reduced kidney function. Furthermore, the use of prescription opioid drugs is reported to induce albuminuria (Barbosa-Leiker et al, 2016), and stimulation of opioid receptors (ORs) was also associated with altered blood pressure (May et al, 1988; Masoudkabir et al, 2013; Mallappallil et al, 2017). Opioid overdose is a huge problem in pain management, especially in patients with CKD. According to the Centers for Disease Control and Prevention, almost 70% of more than 67,000 drug overdose deaths in 2018 involved an opioid. Because more than 17% of Americans had at least one opioid prescription filled, the need for improvements in pain management, especially in those with CKD and hypertension, has risen.

The role of opioid-mediated signaling pathways in podocytes, an essential cellular component of the glomerular filtration and albumin permeability barrier, has not yet been thoroughly investigated. Podocyte damage can be triggered by excessive calcium influx, which leads to glomerular disease and frequently occurs with hypertension. Intracellular calcium ([Ca$^{2+}$]$_i$) influx in podocytes can be mediated by different G protein–coupled receptor (GPCR) pathways targeting members of the transient receptor potential cation (TRPC) channel family, including TRPC6 (Ilatovskaya & Staruschenko, 2015; Ilatovskaya et al, 2017; Dryer et al, 2019; Hall et al, 2019). We hypothesize that stimulation of ORs contributes to dysregulation of [Ca$^{2+}$]$_i$ homeostasis in podocytes, aggravating the development of renal damage in hypertensive conditions. In the present study, we investigated several aspects of OR signaling in podocytes; namely, we identified which ORs (μ-, δ-, or κ-) cause changes to [Ca$^{2+}$]$_i$ levels, assessed the effect of chronic opioid treatment on blood pressure and renal function during salt-induced hypertension, and established TRPC6 as a downstream effector of κ-OR signaling. To do this, we tested the effect of specific

---

[1]Department of Physiology, Medical College of Wisconsin, Milwaukee, WI, USA   [2]Cardiovascular Center, Medical College of Wisconsin, Milwaukee, WI, USA   [3]Division of Nephrology, Department of Medicine, Medical College of Wisconsin, Milwaukee, WI, USA   [4]Clement J. Zablocki VA Medical Center, Milwaukee, WI, USA

Correspondence: staruschenko@mcw.edu
*Daria Golosova and Oleg Palygin contributed equally to this work

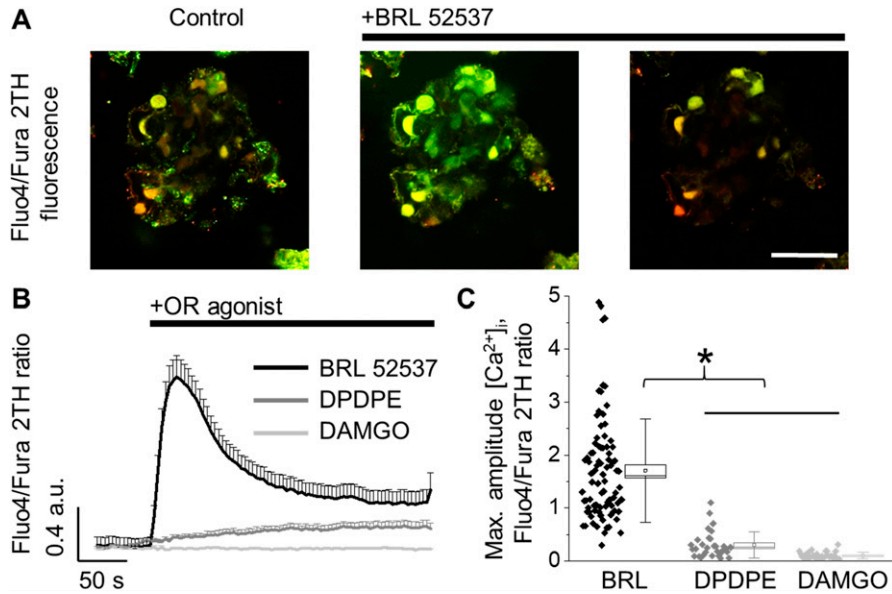

**Figure 1. $[Ca^{2+}]_i$ flux in response to application of opioid receptor (OR) agonists in podocytes from freshly isolated rat glomeruli.**
**(A)** Representative image of a glomerulus loaded with ratiometric calcium dyes (Fluo-4/Fura 2TH; merged) before and after the application of specific $\kappa$-OR agonist (100 $\mu$M, BRL 52537). Scale bar is 50 $\mu$m. **(B)** Response of podocytes to different OR agonists ($\kappa$-OR – 100 $\mu$M BRL 52537, $\mu$-OR – 100 $\mu$M DAMGO, or $\delta$-OR – 50 $\mu$M DPDPE). **(C)** Summary for the peak fluorescence intensity following an application of the OR agonists shown at B ($n \geq 30$ podocytes; one-way ANOVA, $P < 0.001$, Tukey's post hoc $P < 0.05$).

OR agonists and TRPC channel antagonists on $[Ca^{2+}]_i$ transport in rat and human podocytes and evaluated the impact of chronic opioid treatment on glomerular damage and blood pressure in Dahl salt-sensitive (SS) rats.

Here, we show, for the first time that stimulation of G protein–coupled $\kappa$-ORs in rats and human kidney tissue leads to a transient elevation in podocyte $[Ca^{2+}]_i$ via $\kappa$-OR/TRPC6 signaling pathway, which is activated during hypertension, and aggravates podocyte and renal damage. Our experiments also demonstrate that chronic treatment with a $\kappa$-OR agonist leads to increased temporal changes in blood pressure and exacerbation of multiple glomerulus-based pathological changes, including albuminuria and nephrinuria. Taken together, these data unveil a novel mechanism involving $\kappa$-ORs and TRPC6 in regulation of $[Ca^{2+}]_i$ homeostasis that could be pharmacologically targeted to abate the development of kidney injury during opioid treatment.

## Results

### Kappa-ORs ($\kappa$-OR), but not $\mu$- or $\delta$-ORs, mediate calcium influx in podocytes of freshly isolated rat glomeruli

To test the contribution of opioid signaling to the calcium influx in podocytes we isolated glomeruli from Dahl SS rats, as described in the Materials and Methods section. Podocytes of freshly isolated glomeruli were loaded with calcium fluorescent indicators (Fluo-4/Fura 2TH) and used to test the contribution of ORs to the calcium homeostasis by confocal microscopy (Fig 1A). These experiments determined that application of a specific agonist of $\kappa$-OR (BRL 52537; 100 $\mu$M) resulted in a strong and fast increase in $[Ca^{2+}]_i$. In contrast, application of specific agonists for $\mu$- or $\delta$-OR (DAMGO or DPDPE; 100 and 50 $\mu$M, respectively) had little to no effect (Fig 1B and C). Although application of DPDE revealed a slight increase in calcium transient, the magnitude and kinetics of the response are substantially lower than BRL-induced calcium flux.

### $\kappa$-opioid–mediated calcium signaling requires ionotropic calcium entry through TRPC6 channels in human immortalized podocytes

The response to $\kappa$-OR activation was further tested in cultured human immortalized podocytes. Similar to podocytes from freshly isolated rat glomeruli, human cells exhibited robust $[Ca^{2+}]_i$ transients in response to BRL application (Fig 2A). The effect of BRL application was observed when 2 mM $CaCl_2$ extracellular bath solution was used. The amplitude of $\kappa$-OR–mediated $[Ca^{2+}]_i$ transient was dramatically decreased when the extracellular calcium was 0 mM, indicating the direct involvement of plasma membrane calcium-permeable ion channels (Fig 2A and B). We also determined that magnitude of cytosolic calcium elevation in response to $\kappa$-OR–specific agonist demonstrated a dose–response relationship in the presence of extracellular calcium. The calculated half-maximal effective concentration of BRL ($EC_{50} = 162 \pm 2$ $\mu$M) and typical dose-dependent $[Ca^{2+}]_i$ transients are shown in Fig 2C and D. To ensure specificity of the signal, podocytes were preincubated for 5 min with the $\kappa$-OR antagonist norBNI. In these conditions application of BRL had no effect on $[Ca^{2+}]_i$ (Fig 2E). As we and others described previously, G-protein–coupled signaling in podocytes mainly contributes to changes in $[Ca^{2+}]_i$ through the activation of TRPC6 channels (Dryer et al, 2019; Eckel et al, 2011; Ilatovskaya et al, 2014; Wang et al, 2015). Pre-application of SAR7334 (100 nM), a specific TRPC6 channel inhibitor (Maier et al, 2015; Ilatovskaya et al, 2017), significantly attenuated the effect of $\kappa$-OR–mediated calcium influx in cultured podocytes, reducing the amplitude of the signal to the levels observed in 0 mM $[Ca^{2+}]_{out}$ (Fig 2E). These experiments support the idea that ionotropic extracellular calcium influx into the podocytes occur via a $\kappa$-OR/TRPC6 pathway.

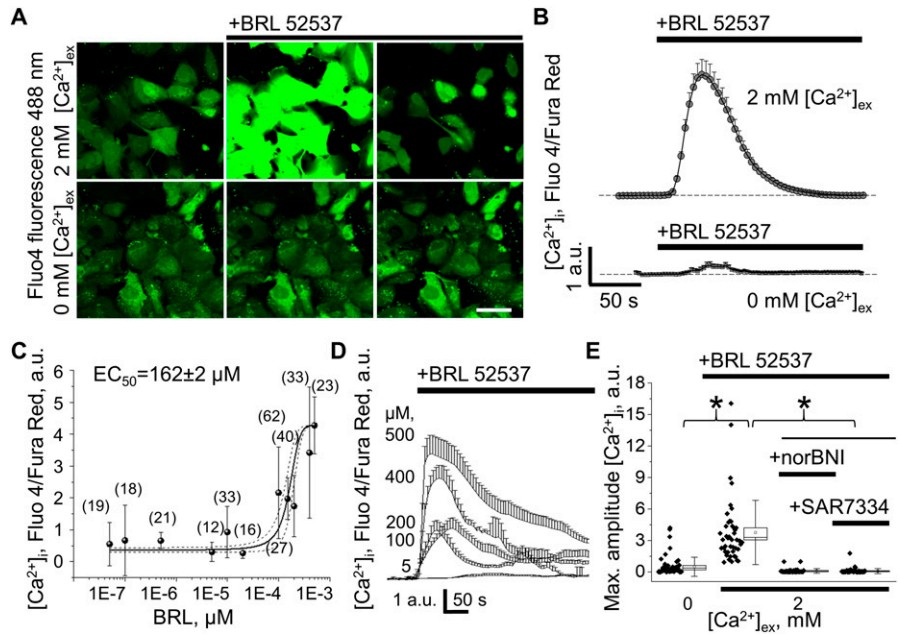

**Figure 2. κ-opioid receptor (OR) activation causes [Ca²⁺]ᵢ mobilization via TRPC6 channels in human immortalized podocytes.**
**(A)** Representative example of specific κ-OR agonist (100 μM, BRL 52537) application in the presence or absence of extracellular calcium (Fluo4 AM fluorescence). Scale bar is 50 μm. Images shown in Fig 2A are taken at 1, 1.5, and 6 min from the start of experiments either in 0 or 2 mM [Ca²⁺]ₑₓ (bottom and top images, respectively). The source data videos are available; 1 min of experiments equals to ~2 s in the videos. **(B)** Mean [Ca²⁺]ᵢ transient in human podocytes after κ-OR stimulation in 2 or 0 mM calcium-containing bath solution (top and bottom traces, respectively). **(C)** Dose–response plot for peak [Ca²⁺]ᵢ increase induced by BRL 52537 (2 mM [Ca²⁺]ₑₓ) in human podocyte culture (a number of podocytes analyzed per point on the graph are shown). **(D)** Representative traces show dose-dependent changes in [Ca²⁺]ᵢ after applications of BRL 52537. **(B, E)** Peak amplitude of the [Ca²⁺]ᵢ after κ-OR agonist stimulation in experiments shown in (B), or during a preincubation of human podocytes with κ-OR antagonist (norBNI) or TRPC6 inhibitor (100 nM, SAR7334) in 2 mM [Ca²⁺]ₑₓ (n ≥ 38 cells; one-way ANOVA, P < 0.001, Tukey's post hoc P < 0.05, all pairwise k = 6).
Source data are available online for this figure.

## Podocytes from freshly isolated human glomeruli express κ-ORs and exhibit κ-OR/TRPC6–mediated calcium influx

Glomeruli were isolated from discarded human kidney transplants utilizing recently described by us the vibrodissociation method, which further allows assessment of the functional cell properties by confocal or electrophysiological techniques (Isaeva et al, 2019). The human glomeruli were used to delineate the functional role of the κ-OR/TRPC6 pathway in the opioid-mediated calcium response. We demonstrated that acute application of the κ-OR agonist, BRL, results in [Ca²⁺]ᵢ transients, which are similar to those seen in podocytes from freshly isolated rat glomeruli and cultured human podocytes, thus validating our choice of models to study this signaling pathway. Moreover, the BRL-induced calcium response was abolished by a TRPC6 specific inhibitor, SAR7334, which is consistent with our data obtained in cultured human or rat glomeruli podocytes (Fig 3A and B). Podocyte-specific localization of κ-ORs in freshly isolated human tissue was verified through immunohistochemical staining of κ-OR protein and the podocyte-

specific protein, podocin (Fig S1). The interaction and co-localization of TRPC6 with disease-associated slit diaphragm proteins (nephrin, podocin, and CD2AP) was described (Reiser et al, 2005; Farmer et al, 2019). Thus, κ-ORs can directly interact with TRPC6 in podocytes.

## Kappa-opioid stimulation induces podocyte cell shape changes via actin cytoskeleton remodeling

[Ca²⁺]ᵢ dysregulation in the pathogenesis of podocytopathy is directly correlated with the compromised integrity of the glomerular filtration barrier (GFB) and podocyte foot processes effacement (Greka & Mundel, 2011). TRPC6 channels are known to modulate the actin cytoskeleton via calcium-dependent Rho GTPase signaling (Hall et al, 2019; Wang et al, 2020; Yu et al, 2020). To evaluate dynamic changes of the podocyte cytoskeleton mediated by κ-OR/TRPC6 pathway, we used confocal microscopy to evaluate actin cytoskeleton structure changes following BRL stimulation in fixed cultured human podocytes treated with a fluorophore-conjugated high-affinity F-actin probe (rhodamine phalloidin). Stimulation of κ-ORs with BRL

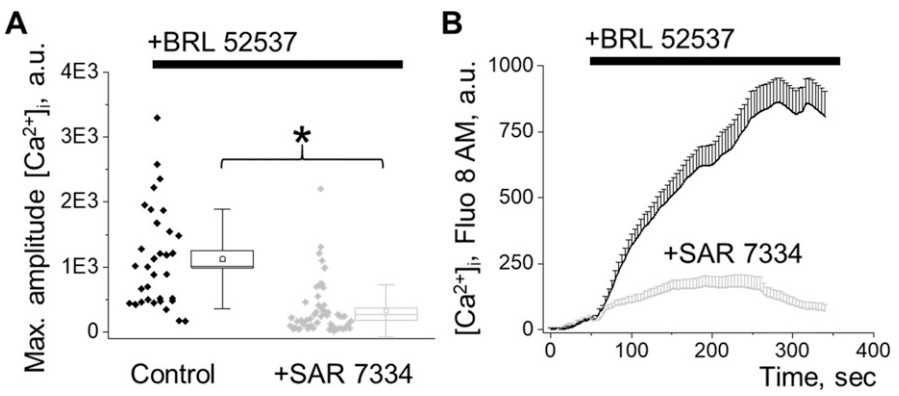

**Figure 3. Assessment of cytosolic calcium response to κ-opioid receptor (OR)/TRPC6 pathway activation in podocytes of the freshly isolated human glomerulus.**
**(A)** Peak amplitude of the [Ca²⁺]ᵢ transient mediated by κ-OR agonist stimulation shown in (B) (n ≥ 33 cells; ANOVA, P < 0.001). **(B)** [Ca²⁺]ᵢ response in podocytes from freshly isolated human glomeruli exposed to κ-OR agonist (100 μM, BRL 52537) under control or TRPC6 blockade (50 μM, SAR7334 preincubation) conditions.

promoted a visible decrease in podocyte surface area (Fig 4A). BRL-treatment also resulted in actin-network structural reorganization, transitioning from more regularly distributed actin filaments to more sporadic actin filament bundles (Fig 4B). To further visualize and quantify changes in podocyte foot process dynamics during $\kappa$-OR stimulation, we applied a scanning ion-conductance microscope (SICM) topographic imaging approach (Fig 4C). The 3D topographical images allowed detection of normal lamellipodial protrusions in live cells under control conditions followed by a strong retraction pattern mediated by $\kappa$-OR stimulation (Fig 4D and Video 1). The cytoskeletal dynamics and structural plasticity of podocytes are essential for efficient glomerular filtration and proper renal function. Thus, the observed phenomena may suggest a negative impact of opioids to the formation, repair, and dynamic of the kidney filtration barrier.

### Chronic treatment with $\kappa$-OR agonist aggravates salt-induced hypertension and exacerbates podocyte damage in Dahl SS rats

Chronic elevation of $[Ca^{2+}]_i$ is strongly associated with progressive glomerular disease and decline in renal function. To further explore the impact of chronic opioid treatment, we used the Dahl SS rat, which is an established animal model for SS hypertension and kidney injury. The dietary sodium content was changed from NS to HS (0.4% and 8% NaCl, respectively), which causes profound elevation of blood pressure and is accompanied by a rapid development of renal damage. To test the contribution of $\kappa$-ORs, HS-treated animals were administered 1.5 mg/kg i.v. bolus injections of $\kappa$-OR agonist or vehicle during the development of hypertension (Fig 5A). Chronic opioid treatment induced significant increases in mean arterial pressure at the late stage of hypertension (Fig 5B), which corresponds to the period of progressive decline in glomerular filtration rate in this model (Cowley et al, 2013). The BRL-induced elevation in high blood pressure was accompanied by detectable changes in electrolyte homeostasis. As shown in Tables 1 and 2, analyses of both urinary and plasma electrolyte levels indicated significant changes in electrolyte transport during chronic opioid exposure (19% ± 1% versus 13% ± 2%, 10% ± 1% versus 6% ± 1% and 8% ± 1% versus 4% ± 1% for $FE_K$, $FE_{Cl}$, and $FE_{Na}$, correspondingly, ANOVA, $P < 0.05$). Opioids, including $\kappa$-OR agonists, produce changes in urine output and urine sodium excretion by multiple integrated neural and hormonal mechanisms within the periphery, central nervous system, and kidneys (Kapusta, 1995). Several experimental data provide evidence for the role of central $\kappa$-ORs mechanisms in the regulation of urinary sodium and water excretion involving tubular events and altered water homeostasis, although the exact mechanism remains unclear (Ashton et al, 1990; Mercadante & Arcuri, 2004). The negative impact of BRL was also reflected by increased urinary albumin excretion (Fig 5C). Opioid-induced glomerular injury was elevated but did not reach statistical difference between the two groups (Fig S2). Although both hypertensive groups have substantial glomerular injuries, the probability of finding a glomerulus with a score of three or four was higher in BRL-treated rats, as seen by the rightward shift in the cumulative probability distribution. Importantly, basal $[Ca^{2+}]_i$ concentrations in the podocytes were elevated in the opioid-treated rats on a HS diet (Fig 5D). Moreover, BRL-treated rats had a significant increase in podocyte damage, as demonstrated by the substantial urinary nephrin shedding (Fig 5E). Urinary nephrin shedding indicates loss of the glomerular slit diaphragm, which also correlates with the increased albuminuria in BRL-treated rats. Furthermore, Western blot analysis of opioid- and vehicle-treated groups revealed a significant elevation in pro-caspase 3, a marker of apoptotic injury. Pro-caspase 3 level was also upregulated following a HS diet, with an additional trend toward further increase in the BRL-treated group (Fig S3).

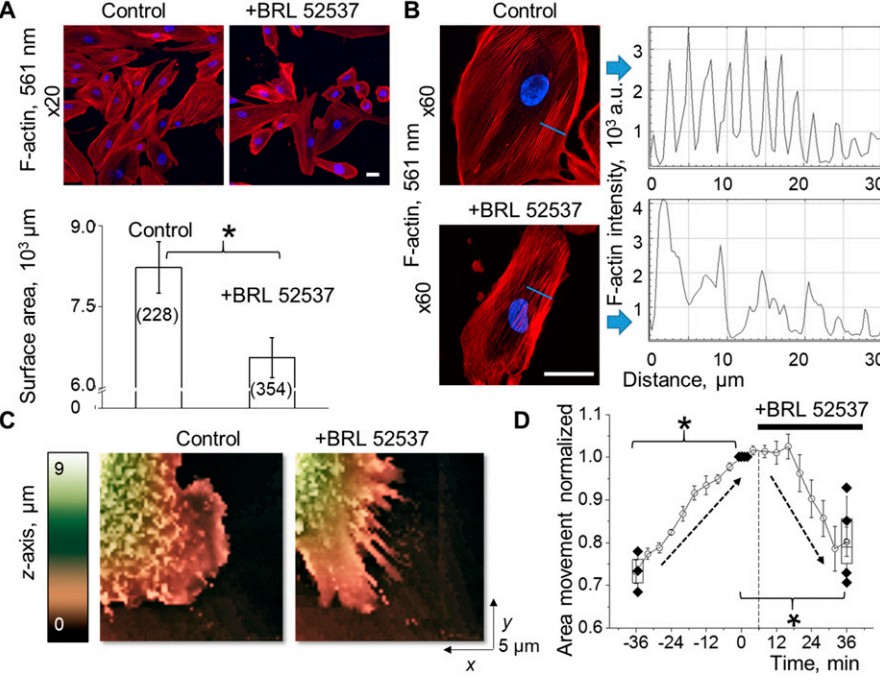

**Figure 4. $\kappa$-opioid receptor (OR) agonist induces podocyte cell shape changes via the actin cytoskeleton remodeling.**
**(A)** An application of $\kappa$-OR agonist to human-cultured podocytes promotes significant shrinkage in the cell surface area (top panel, scale bar is 50 $\mu$m). Changes in the mean cell surface area were obtained by comparisons between BRL-treated and control cells (number of cells is indicated in the brackets, ANOVA, $P < 0.001$; panel below). **(B)** Following BRL-induced podocyte shrinkage, the actin cytoskeleton pattern shows reinforcement and formation of the actin stress fibers. The reorganization of the F-actin cytoskeleton is indicated by the changes in the rhodamine phalloidin staining pattern through the blue line. Scale bar is 50 $\mu$m. **(C)** Surface topography 3D images obtained by scanning ion-conductance microscopy shows lamellipodia dynamics in human podocytes treated with $\kappa$-ORs agonist (changes in z-axis indicated by pseudocolor). **(D)** Summary of scanning ion-conductance microscope topography indicates normal lamellipodial protrusion under control conditions followed by a strong retraction pattern mediated by $\kappa$-OR stimulation (n ≥ 3 cells; ANOVA, $P < 0.002$, Tukey's post hoc $P < 0.05$).

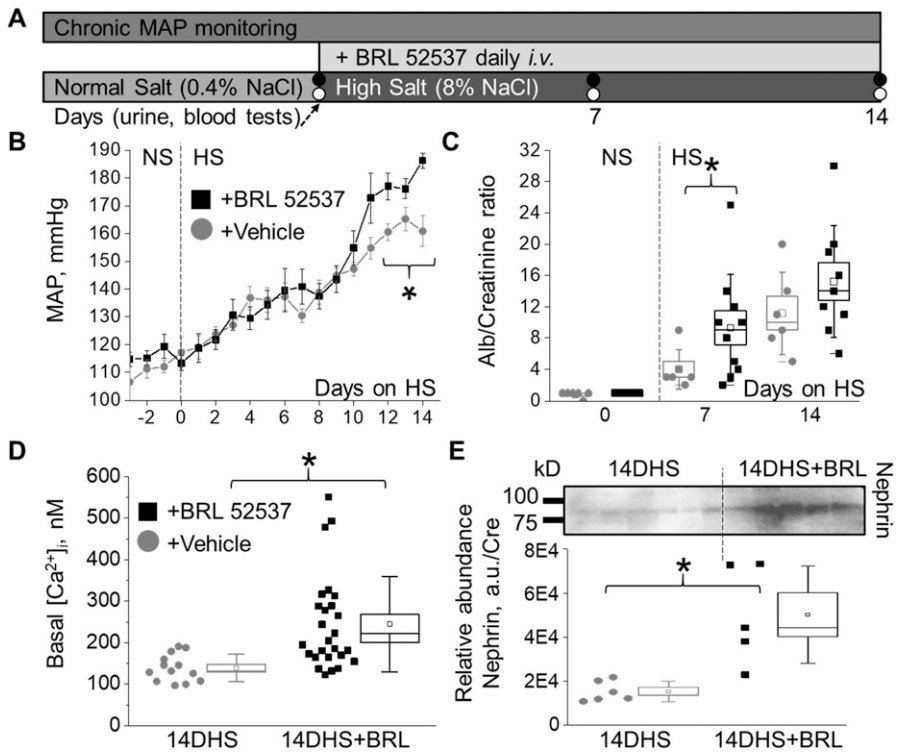

**Figure 5. Chronic treatment with κ-opioid receptor (OR) agonist aggravates salt-induced hypertension and podocyte damage in Dahl salt-sensitive (SS) rats.** **(A)** Schematic representation of the experimental protocol. **(B)** The development of mean arterial pressure in SS rats after changing the diet from a normal salt (NS, 0.4%; day 0) to a HS (8% NaCl) and chronic treatment with κ-OR agonist or vehicle (n ≥ 5 rat per group; ANOVA, $P < 0.05$). **(C)** Urinary albumin (Alb/Cre, 24 h collection) changes for the experiments shown in (B). **(D)** Basal calcium levels in podocytes of freshly isolated glomeruli from hypertensive SS rats chronically exposed to κ-OR agonist or vehicle (n ≥ 13 cells; ANOVA, $P < 0.003$). **(E)** Western blot analysis of nephrin in the urine samples (24-h collection) from hypertensive (2 wk on HS) and BRL-treated rats (n = 5 rats per group; ANOVA, $P < 0.005$).

## TRPC6 expression is elevated under hypertensive conditions and promotes pathological increase in $Ca^{2+}$ influx in response to κ-OR stimulation

Western blot analysis revealed that TRPC6 expression was elevated in the hypertensive animal group (Fig 6A and B) and trended to be higher in the hypertensive opioid-treated group, although it did not reach statistical significance. This was also reflected in the RT-qPCR analysis, where we observed similar trends (Fig S4). As a potential explanation of this phenomenon, the increased podocyte damage demonstrated by the increased nephrinuria (Fig 5E), results in fewer podocytes in the hypertensive BRL-treated group; therefore, less TRPC6 expression was due to reduced cell number rather than reduced protein expression per se.

Importantly, when we examined the acute podocyte response to κ-OR stimulation in glomeruli from rats on NS or HS diets, we found substantial changes in podocyte calcium homeostasis between normotensive and hypertensive groups. As shown in Fig 6C, application of BRL led to a sustained elevation of podocyte $[Ca^{2+}]_i$ in hypertensive animals, which most likely represented a substantial dysregulation of $[Ca^{2+}]_i$ homeostasis observed under HS conditions. Therefore, this study revealed that, in hypertension, opioid treatment triggers excessive calcium influx in podocytes, further promoting cell apoptosis and kidney damage (Fig 7).

## Discussion

Despite the common use of opioid analgesics for pain relief in patients with renal insufficiency and hypertension, there is limited

evidence about the factors underlying exacerbated cardiorenal pathologies associated with opioid toxicity. Previous studies have shown that ORs are expressed in the kidney and particularly in podocyte (Lan et al, 2013). Podocytes play a key role in the maintenance of glomerular structure and control permeability. Podocyte

**Table 1. Blood electrolytes analysis of salt-sensitive rats during chronic experiment with or without treatment with the opioids within 14 d on HS.**

|  | NS | NS |
|---|---|---|
| Potassium | 3.2 ± 0.1 | 3.4 ± 0.1 |
| Sodium | 143.8 ± 0.4 | 141.4 ± 0.4 |
| Calcium | 1.35 ± 0.01 | 1.32 ± 0.01 |
| Chloride | 113 ± 4 | 106 ± 2 |
|  | **7DHS+Vehicle** | **7DHS+BRL** |
| Potassium | 3.2 ± 0.1 | 3.1 ± 0.2 |
| Sodium | **146 ± 1** | **150 ± 2*** |
| Calcium | 1.38 ± 0.01 | 1.30 ± 0.02 |
| Chloride | **104 ± 1** | **110 ± 2*** |
|  | **14DHS+Vehicle** | **14DHS+BRL** |
| Potassium | 3.1 ± 0.1 | 3.0 ± 0.1 |
| Sodium | 143.3 ± 0.5 | 146.7 ± 1.6 |
| Calcium | **1.36 ± 0.01** | **1.23 ± 0.04*** |
| Chloride | 101 ± 1 | 106 ± 2 |

κ-OR agonist treated rats (1.5 mg/kg BRL 52537). NS (0.4% NaCl); 7DHS, 14DHS, 7 d and 14 d on HS (8% NaCl), respectively (n ≥ 5 rat per group; $t$ test *$P < 0.05$, BRL- versus vehicle-treated rats on the same day of treatment).

**Table 2.  Urine electrolytes/creatinine ratio in salt-sensitive rats during chronic opioid treatment within 14 d on HS.**

| | NS | NS |
|---|---|---|
| Potassium | 8.5 ± 2 | 15 ± 2 |
| Sodium | 33 ± 5 | 23 ± 2 |
| Calcium | 0.24 ± 0.04 | 0.26 ± 0.03 |
| Chloride | 44 ± 8 | 32 ± 3 |
| | **7DHS+Vehicle** | **7DHS+BRL** |
| Potassium | 19 ± 3 | 16 ± 1 |
| Sodium | **437 ± 41** | **327 ± 17***  |
| Calcium | 3.2 ± 0.5 | 2.9 ± 0.2 |
| Chloride | **402 ± 34** | **310 ± 15***  |
| | **14DHS+Vehicle** | **14DHS+BRL** |
| Potassium | **19 ± 1** | **15 ± 1***  |
| Sodium | **372 ± 30** | **255 ± 9***  |
| Calcium | **2.9 ± 0.2** | **2.1 ± 0.1***  |
| Chloride | **323 ± 24** | **241 ± 9***  |

κ-OR agonist treated rats (1.5 mg/kg BRL 52537). NS (0.4% NaCl); 7DHS, 14DHS − 7 d and 14 d on HS (8% NaCl), respectively (n ≥ 5 rat per group; t test *P < 0.05, BRL- versus vehicle-treated rats on the same day of treatment).

injury is the common pathological process in many glomerular diseases such as minimal change disease, membranous glomerulopathy, focal segmental glomerulosclerosis (FSGS), diabetic nephropathy, and lupus nephritis. Several studies have indicated that mice treated with morphine develop podocyte injuries and increased urinary albumin excretion (Lan et al, 2013) as well as podocyte foot process effacement accompanied by the loss of GFB integrity (Weber et al, 2012).

Calcium signaling plays an essential role in podocyte pathophysiology (Dryer et al, 2019; Hall et al, 2019; Lu et al, 2019). TRPC

proteins, belonging to the larger TRP superfamily, form $Ca^{2+}$-permeable channels and play an important role in the pathogenesis of renal and cardiovascular diseases (Abramowitz & Birnbaumer, 2009; Spires et al, 2019). Several independent laboratories have reported the identification of TRPC6 mutations associated with the autosomal dominant form of FSGS (Reiser et al, 2005; Winn et al, 2005; Heeringa et al, 2009; Kim et al, 2018). These genetic discoveries highlight the critical importance of the podocyte and the TRPC6 channel, particularly in the maintenance of the GFB (Ilatovskaya & Staruschenko, 2015). TRPC activation is dependent on various GPCR-signaling mechanisms, including metabotropic purinergic P2Y and Ang II receptors in podocyte (Anderson et al., 2014; Ilatovskaya et al., 2014; Roshanravan and Dryer, 2014; Wang et al., 2015). Interestingly, it was shown that stimulation of ORs was able to stimulate TRPC channels and mediate corresponding $Ca^{2+}$ mobilization in HEK293 cell line (Miller et al, 2011). The presence of ORs in podocyte could be an additional factor in the stimulation of GPCR/TPRC6 calcium cell entry, which may contribute to pathological calcium signaling. In particular, in salt-induced hypertension, TRPC6 overexpression and overstimulation via the κ-OR/TRPC6 pathway may lead to the loss of podocytes and aggravate renal damage. There is a common consensus that TRPC channels are potentiated by tyrosine kinase receptor-mediated activation of phospholipase C or GPCRs. Subsequently, phosphatidylinositol 4,5-bisphosphate is cleaved, thus liberating diacylglycerol and inositol 1,4,5-trisphosphate (Clapham, 2003). GPCRs coupled to Gq signaling activate TRPC6, suggesting that Gq-dependent TRPC6 activation underlies glomerular diseases. The important role of TRPC6 in this context was highlighted in a mouse model with constitutively active $Gq_{\alpha}$ subunits in podocytes (Wang et al, 2015). In our experiments a pre-application of SAR7334 diminished BRL-induced calcium entry; as such, we may speculate that κ-opioid metabotropic receptors are positively coupled to TRPC6 activity; however, contributions of other ionotropic channels may also be involved (Ilatovskaya et al, 2014). Taken

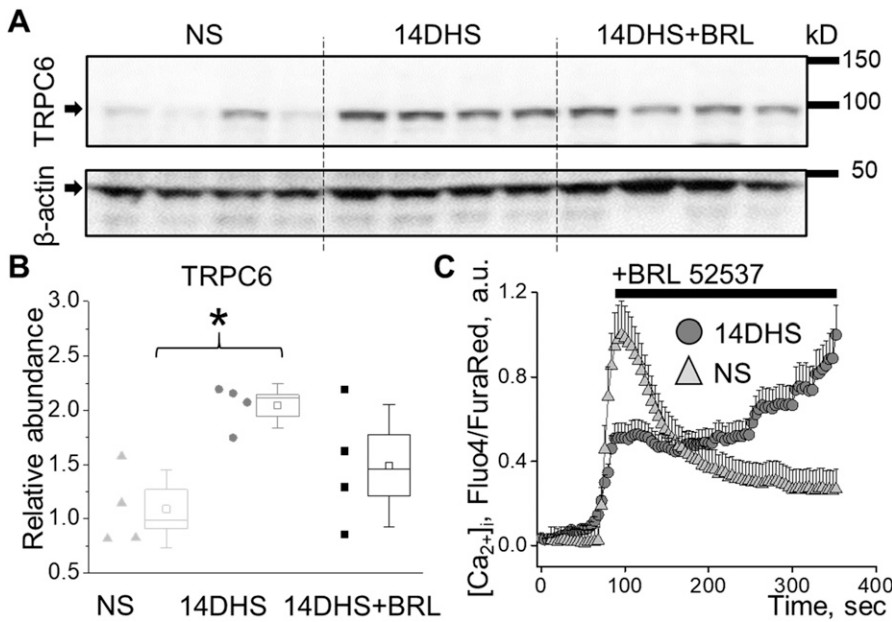

**Figure 6.  Increased TRPC6 expression leads to exacerbated podocyte apoptosis in response to κ-OR stimulation in hypertension.**
**(A)** Western blot analyses of the TRPC6 expression in the kidney cortex of normotensive (NS), hypertensive (14DHS), and hypertensive treated with BRL (14DHS+BRL) groups. **(B)** Summary graph of TRPC6 expression as shown in A (n = 4 rats per group, ANOVA, P < 0.03, Tukey's post hoc P < 0.05). **(C)** Mean $[Ca^{2+}]_i$ transient in podocytes of glomeruli freshly isolated from salt-sensitive rats fed normal salt (NS) or after 2 wk on HS (14DHS). Note the sustained $[Ca^{2+}]_i$ response to acute stimulation of κ-ORs in hypertensive animals.

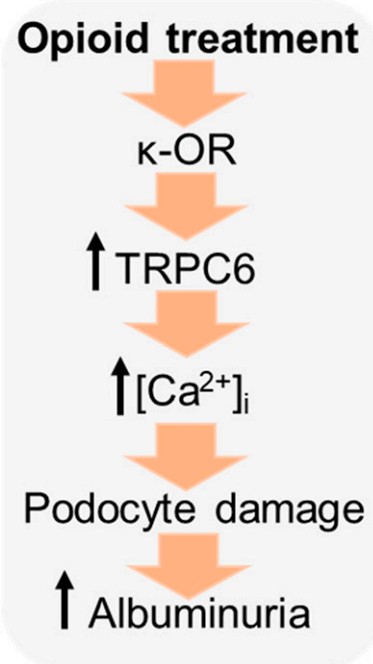

**Figure 7. Schematic demonstrating the contribution of the κ-OR/TRPC6 pathway to the pathological elevation in [Ca²⁺]ᵢ, and consecutive podocyte foot processes retraction, apoptosis, and progression of renal damage in hypertension during opioid treatment.**

together, these aspects point to specific involvement of the κ-OR pathway in renal damage through an effect on podocyte and reveal a potential link to mechanisms of opioid-induced damage in CKD and renal failure (Barbosa-Leiker et al, 2016; Novick et al, 2016; Mallappallil et al, 2017).

Our studies demonstrate the functional presence of κ-ORs, but not μ-ORs or δ-ORs, in podocytes. The stimulation of κ-ORs with a specific agonist shows rapid and transient mobilization of $[Ca^{2+}]_i$ in freshly isolated rat glomeruli or human cultured podocytes. This response could be blocked by a κ-OR antagonist or by inhibition of TRPC6-mediated $Ca^{2+}$ entry. The activation of the κ-OR/TRPC6 pathway also resulted in podocyte cell shape changes and remodeling of the actin cytoskeleton. Thus, overstimulation of κ-ORs during chronic opioid exposure may result in disorganized F-actin arrangement and foot processes effacement in podocytes. In addition, we were able to confirm the presence of κ-OR/TRPC6 signaling in podocytes of freshly isolated human glomeruli, using a recently developed protocol for the isolation of nephron segments and glomeruli from discarded human kidney transplants (Isaeva et al, 2019).

Although opioids are commonly prescribed analgesics for hypertensive patients, it was established that the use of these drugs could worsen an overall outcome of cardiovascular diseases: 61% of patients prescribed opioids suffer from elevated arterial blood pressure and 15% have renal failure (Herzig et al, 2014). The risk of renal damage due to long-term use of opioids has been established (Weber et al, 2012; Lan et al, 2013; Kimmel et al, 2017; Sethi, 2018). Our studies revealed that chronic treatment of hypertensive rats with

κ-OR agonist BRL accelerated podocyte damage and further exacerbated proteinuria and kidney injury. Indeed, acute κ-OR activation in podocytes of freshly isolated glomeruli from hypertensive animals led to a sustained elevation of $[Ca^{2+}]_i$. This large and sustained calcium influx into cells can initiate apoptotic signaling cascades (Trump & Berezesky, 1996), which was also reflected by elevated basal $[Ca^{2+}]_i$ in podocytes and increased urinary nephrin shedding. The overexpression of TRPC6 channels during the development of hypertension provided an additional source for $Ca^{2+}$ entry and triggered an excessive $Ca^{2+}$ influx through the κ-OR/TRPC6 pathway in the case of opioid treatment. After opioid-induced podocyte injury, the existing podocytes sustained additional injury as a consequence of overexpression of TRPC6 channels in hypertension, which forms a vicious circle. When albuminuria precedes elevated blood pressure, we may conclude that opioids directly damage podocytes, revealing their role as an important cause and consequence of CKD. Although there are potential limitations of our studies including neurological (Chen et al, 1991) and cardiovascular effects (Peart et al, 2004) of BRL, the in vitro and ex vivo experiments showed the direct impact of opioid treatment to podocyte damage. In conclusion, the present work represents the first evidence of the role of κ-OR/TRPC6 signaling in the control of calcium handling in podocytes and reveals the mechanism by which opioids exacerbate kidney damage in hypertension. These findings are important for further understanding of the role of ORs in podocyte damage and for advancing our knowledge of the pathogenesis of hypertension and CKD in the context of pain management.

# Materials and Methods

### Experimental protocol and animals

Animal use and welfare adhered to the National Institutes of Health Guide for the Care and Use of Laboratory Animals, following protocols reviewed and approved by the Medical College of Wisconsin Institutional Animal Care and Use Committee. 8-wk-old male Dahl salt-sensitive rats (SS; SS/JrHsdMcwi) rats were provided normal (0.4% NaCl, # D113755; Dyets Inc.; NS) and high salt (8% NaCl, # D100078; Dyets Inc.; HS) diets, and water ad libitum. At the age of 7.5 wk, the rats were anesthetized with inhalation of 2.5% isoflurane in 0.5 liters/min $O_2/N_2$ (30/70) flow, and catheters (#RPT080) were implanted in the femoral artery and vein, tunneled subcutaneously, and exteriorized at the back of the neck in a lightweight tethering spring. After recovery from anesthesia, all rats were placed into individual cages. The catheters were connected to a pressure transducer (#041516504A; Argon Medical Devices) via swivels (#375/D/22; Instech) for arterial blood pressure acquisition and daily intravenous bolus drug infusion. The rats were allowed to recover for at least 3 d after surgery. κ-OR agonist (BRL 52537 hydrochloride, #0699; Tocris Bioscience) was used at a dose of 1.5 mg/kg, as previously reported (Cheng et al, 2007). Urine and blood samples were collected on day 0 (NS, no treatment), day 7 and 14 on HS. Treatment with BRL 52537 was initiated from the first day of the HS challenge.

## Cell culture

An immortalized human podocyte cell line was provided by Dr. Moin Saleem (Children's Renal Unit and Academic Renal Unit, University of Bristol, Southmead Hospital). The cells were cultured as described previously (Ni et al, 2012). Briefly, the cells were grown at 33°C with 5% $CO_2$ in an RPMI-1640 medium (#11875093) supplemented with insulin–transferrin–selenium (#41400045) and 10% FBS (#10437028; Thermo Fisher Scientific, Inc.). Human podocytes between passages 4–15 were used. To induce differentiation, the cells were switched to 37°C with 5% $CO_2$ for 10–14 d. Cells were then incubated in serum-free media for 3–5 h before experimental use.

## Kidney isolation, histological staining, and analysis

Rats were anesthetized, and their kidneys were flushed (3 ml/min/kidney until blanched) with PBS via aortic catheterization. For each rat, one kidney was used for glomeruli isolation, and the other kidney was used for Western blot and immunohistochemistry analyses. Rat kidneys were formalin fixed, paraffin embedded, sectioned, and mounted on slides (Palygin et al, 2017). Slides were stained with Masson's trichrome stain. A double-blind glomerular injury score was assessed using a 0–4 scale, as previously described (Palygin et al, 2019). Immunohistochemical staining of κ-OR protein in human kidneys was visualized with κ-OR-1 antibody (KOR-1 (D-8), 1:50, #sc-374479; Santa Cruz Biotechnologies).

Human kidneys were obtained from the organ procurement organization, perfused with preservation solution, and stored on ice before the glomeruli isolation. Our previous studies revealed that these kidneys are viable and functional in vitro (Isaeva et al, 2019). Glomeruli were isolated by the vibrodissociation or sieving methods, as described previously (Ilatovskaya et al, 2018; Spires et al, 2018; Isaeva et al, 2019).

## Western blotting and immunofluorescence

Changes in protein expression in renal cortex homogenates or urine samples were assessed using primary antibodies (caspase 3, 1:1,000, #PA5-77887; Thermo Fisher Scientific; TRPC6, 1:200, #ACC-017; Alomone; nephrin, 1:500, #ab58968; Abcam). The membranes were blocked with 2% BSA in TBS and 0.01% Tween 20 overnight at room temperature and then incubated with primary antibody overnight at room temperature. The secondary antibody (1:10,000) was diluted in 2% BSA in TBS and 0.01% Tween 20, and membranes were incubated at room temperature for 1 h. Protein loading was assessed by immunoblotting using rabbit anti-actin antibodies (1:1,000; Cell Signaling Technology).

For Podocin (NPHS2) or κ-OR immunofluorescence labeling, freshly isolated human glomeruli were allowed to adhere to microscopy coverslips double-coated with poly-L-lysine. Adhered glomeruli were fixed with chilled 4% paraformaldehyde in PBS with 1 mM $CaCl_2$ and 2 mM $MgCl_2$ for 20 min, then gently washed 3× with ice-cold PBS. Next, the glomeruli were probed with Podocin and κ-OR-1 primary antibody in a 2% BSA + 0.1% Triton X-100 PBS solution overnight at 4°C (NPHS2, 1:250, #Ab50339; Abcam; KOR-1 (D-8), 1:50, #sc-374479; Santa Cruz Biotechnologies). The following day, glomeruli were washed three times with cold PBS and incubated

with Alexa 488 and Alexa 594 fluorophore–labeled secondary antibodies (1:500; Thermo Fisher Scientific) in a 2% BSA-PBS solution at room temperature in the dark. After three PBS washes, glomeruli were incubated with 0.5 μg/ml Hoescht nuclear stain in PBS for 10 min at room temperature in the dark. After five final washes with PBS, tissue was preserved and coverslip-mounted with Fluoromount-G (SouthernBiotech). Images were captured on a confocal Nikon A1R inverted microscope using a Plan Apo 40x/NA 0.95 controlled by Nikon Elements AR software (Nikon). Post-image processing was performed with Fiji (ImageJ; National Institutes of Health) image software.

## RT-qPCR analysis

Kidney tissue was isolated after 14 d of HS challenge with age-matched SS rats on NS diet. Sections of the whole kidney were flash-frozen in liquid nitrogen. Total RNA was extracted using TRIzol Reagent (Thermo Fisher Scientific) according to the manufacturer's protocol. Total RNA quantity was determined by a NanoDrop 2000 spectrophotometer (Thermo Fisher Scientific). To evaluate RNA fragmentation, the size and quantity of RNA transcripts for each sample were assessed using a Pico RNA chip in an Agilent 2000 Bioanalyzer. The expression of selected genes was quantified by RT-qPCR as previously described (Khedr et al, 2019). cDNA from 1 μg of RNA was synthesized using the RevertAid First-Strand cDNA Synthesis Kit (Thermo Fisher Scientific). SYBR Green real-time PCR reactions were carried out on an ABI Prism 7900HT (ABI; Applied Biosystems) using Bullseye EvaGreen qPCR Master Mix (MedSci) according to the manufacturer's directions in 10 μl final volume with samples run in triplicate. The primers (5′–3′) used: TRPC6-2F: forward, tactatcccagcttccggg; reverse, TRPC6-2R ctagcatcttccgcaccact; 18S: reverse, cctgtattgttattttgtcgtcactacct; forward, cggctaccacatccaaggaa. Final Ct values were determined using QuantStudio 6 Flex implemented software and normalized to housekeeping gene 18S. Relative abundance was compared using the $2^{-\Delta\Delta Ct}$ method with the low salt samples considered as control.

## [Ca$^{2+}$]$_i$ imaging and visualization of F-actin with rhodamine phalloidin staining

The laser scanning confocal microscope system Nikon A1-R was used to detect [Ca$^{2+}$]$_i$ transients and actin rearrangements. Samples were imaged using 20× and 60× objective lenses: Plan Apo 20×/NA 0.75 and 60×/NA 1.4 Oil. Open-source software ImiageJ was used for analysis as described previously (Ilatovskaya et al, 2011).

Calcium imaging was performed in time series (xyt, 4 s per frame) with Nikon Elements software. Changes in [Ca$^{2+}$]$_i$ concentration in isolated glomeruli or cultured podocytes were estimated by fluorescent dyes: Fluo 4 (ex. 488 em. 520/20 nm; #20190588; Invitrogen) and Fura Red (ex. 488 em. >600 nm; #21046; AAT Bioquest, Inc). In some experiments Fura Red was replaced with Fura 2-TH (ex. 488 em. >600 nm; #51419; Setareh Biotech). Glomeruli were mounted in poly-l-lysine-covered Mattek dishes and washed for ~10 min with a bath solution containing the following (in Millimolar): 145 NaCl, 4.5 KCl, 2 $CaCl_2$, 2 $MgCl_2$, and 10 Hepes, pH 7.35 (adjusted with NaOH). After the fluorescence signal stabilized, podocytes were identified on the basis of anatomic considerations, and fluorescence intensity

ratios (Fluo-4/Fura Red) were recorded as described previously (Ilatovskaya et al, 2013, 2015). ORs were stimulated with specific agonists (Tocris Bioscience): BRL 52537 ((±)-1-(3,4-dichlorophenyl) acetyl-2-(1-pyrrolidinyl)methylpiperidine hydrochloride, BRL; $\kappa$-OR, #0699), DAMGO ([D-Ala$^2$,NMe-Phe$^4$,Gly-ol$^5$]-enkephalin, 100 $\mu$M, $\mu$-OR; #1171), and DPDPE ([D-Pen$^2$,D-Pen$^5$]-enkephalin, 50 $\mu$M, $\delta$-OR; #1431). Nor-Binaltorphimine dihydrochloride (norBNI, 10 $\mu$M; #0347/10; Tocris Bioscience) and SAR7334 (#5831; Tocris Bioscience) were used to inhibit $\kappa$-OR and TRPC6 channels, respectively. Basal podocyte $[Ca^{2+}]_i$ measurements in rat podocytes were done as described previously (Ilatovskaya et al, 2018).

To detect F-actin cytoskeletal remodeling, immortalized human podocytes were incubated in a 2 mM calcium saline solution (see above) for 5 min and then stimulated with 100 $\mu$M of BRL52537 or vehicle for 5 min. The time frame was chosen based on data obtained from confocal imaging of the calcium transients. Cells were fixed with 4% paraformaldehyde in PBS, treated with 0.1% Triton X-100 in 2% BSA-PBS, and incubated with 2% BSA-PBS containing rhodamine phalloidin (1:500; Sigma-Aldrich) for 15 min, followed by a 0.5 $\mu$g/ml Hoescht nuclear stain in PBS for 10 min. Stained cells were mounted with Vectashield mounting medium (Vector Laboratories). Cells were imaged with the Nikon confocal microscope system described above.

### SICM

The dynamic changes in podocyte cytoskeletal remodeling during opioid treatment were visualized by super-resolution hopping probe ion-conductance microscopy, an advanced three-dimensional topographical ersion of SICM. The identification of podocyte lamellipodial protrusions was carried out using the custom-modified scanner ICNano (ICAPPIC), as previously described (Novak et al, 2009; Palygin et al, 2019). The sample was manually positioned in the x–y directions under an optical microscope, and the scanning pipette was positioned in the z-direction with a piezoelectric actuator. Fine-tipped scanning nanopipettes were pulled from borosilicate glass (outer diameter/internal diameter: 1/0.5 mm) with the horizontal laser puller P-97 (Sutter Instruments). The pipette resistance was in the range of 80–100 M$\Omega$ corresponding to an estimated tip diameter of 90–120 nm. Nanopipettes were held in voltage-clamp mode with an Axopatch 700B patch-clamp amplifier (Axon Instruments). The amplifier headstage was mounted on the z-scanning head. The output signal was monitored by SICM electronics, which simultaneously controlled sample and pipette positioning. The scan system was mounted on a Nikon TE2000-U inverted microscope (Nikon Instruments). Raw data were processed using SICM ImageViewer microscopy analysis software (ICAPPIC).

### Statistics

Data are presented as means ± SE. In the box plot graphs, the ends of the box are ±SE, and the median is marked by a vertical line inside the box. The whiskers are ±SD. Data were tested for normality (Shapiro–Wilk) and equal variance (Levene's homogeneity test). Statistical analysis consisted of one-way ANOVA or student t test as indicated (SigmaPlot 12.5 or OriginPro 9.0), with a P-value of <0.05 considered significant. In addition, when an ANOVA test was significant

($P < 0.05$), Tukey's multiple-comparisons adjustment was applied to all pairwise P-values.

## Supplementary Information

## Acknowledgements

The authors would like to acknowledge the help of Christine Duris and Tanya Bufford (Children's Research Institute Histology Core), Lisa Henderson, Jenifer Phillips, and Camille Taylor (Physiology Biochemistry Core). We also thank Demi Carter (Hampton University) for initial experiments with the actin staining and Anna Williams for helping with the RT-qPCR analyses (Medical College of Wisconsin). Dr. Moin A Saleem (University of Bristol, UK) is greatly appreciated for providing the human podocytes cell line. We would like to acknowledge the support provided the National Heart, Lung, and Blood Institute grant R35 HL135749 (to A Staruschenko), the American Heart Association (20POST35180224 to D Golosova and 17SDG33660149 to O Palygin) and Department of Veteran Affairs grant I01 BX004024 (to A Staruschenko).

### Author Contributions

D Golosova: conceptualization, formal analysis, funding acquisition, investigation, and writing—original draft, review, and editing.
O Palygin: conceptualization, data curation, formal analysis, investigation, methodology, and writing—original draft, review, and editing.
R Bohovyk: formal analysis, investigation, methodology, and writing—review and editing.
CA Klemens: formal analysis, investigation, and writing—review and editing.
V Levchenko: formal analysis, investigation, and writing—review and editing.
DR Spires: formal analysis, investigation, and writing—review and editing.
E Isaeva: formal analysis, investigation, and writing—review and editing.
A El-Meanawy: conceptualization, resources, and writing—review and editing.
A Staruschenko: conceptualization, resources, data curation, supervision, funding acquisition, and writing—original draft, review, and editing.

### Conflict of Interest Statement

The authors declare that they have no conflict of interest.

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
