## [Reviewer comments · Life Science Alliance]

Life Science Alliance

Role of opioid signaling in kidney damage during the development of salt-induced hypertension

Daria Golosova, Oleg Palygin, Ruslan Bohovyk, Christine Klemens, Vladislav Levchenko, Denisha Spires, Elena Isaeva, Ashraf El-Meanawy, and Alexander Staruschenko

DOI: <https://doi.org/10.26508/lsa.202000853>

Corresponding author(s): Alexander Staruschenko, Medical College of Wisconsin

Review Timeline:

Submission Date:	2020-07-20
Editorial Decision:	2020-08-18
Revision Received:	2020-09-20
Editorial Decision:	2020-09-28
Revision Received:	2020-10-01
Accepted:	2020-10-02

Scientific Editor: Shachi Bhatt

Transaction Report:

August 18, 2020

Re: Life Science Alliance manuscript #LSA-2020-00853-T

Dr. Alexander Staruschenko
Medical College of Wisconsin
Physiology
8701 Watertown Plank Road
Milwaukee, WI 53226

Dear Dr. Staruschenko,

Thank you for submitting your manuscript entitled "Role of opioid signaling in kidney damage during the development of salt-induced hypertension" to Life Science Alliance (LSA). The manuscript has been reviewed by the editors and outside referees (reviewer comments below). As you will see, the reviewers were quite enthusiastic about the study and its potential impact, but have raised some concerns that should be addressed prior to further consideration of the manuscript at LSA. Therefore, although we are unable to publish the current version of the manuscript, we would encourage you to submit a revised version that addresses all of the referees' concerns. While we appreciate the detailed review by Reviewer # 2, figuring out whether the enhanced damage in kidney due to the augmented hypertension or the direct acting on the renal cells would not be required for publication at LSA - please do discuss this point in the discussion.

The typical time frame for revisions is three months. Please note that papers are generally considered through only one revision cycle, so strong support from the referees on the revised version is needed for acceptance.

Thank you for this interesting contribution to Life Science Alliance. We are looking forward to receiving your revised manuscript.

Sincerely,

Shachi Bhatt
Executive Editor
Life Science Alliance

B. MANUSCRIPT ORGANIZATION AND FORMATTING:

Reviewer #1 (Comments to the Authors (Required)):

I've read with interest this manuscript from the Alexander Staruschenko's group. The study describes functional expression of kappa opioid receptors in rat and human podocytes and investigates some mechanisms triggered by KOR activation in vitro and in vivo. The authors show that KOR ligand induces Ca²⁺ influx through the TRPC channels in podocytes. A prolonged stimulation results in Ca²⁺ overload and podocyte death which, in turn, compromises the integrity of the glomerular filtration barrier leading to renal damage and loss of blood pressure control. These are significant findings as they provide novel insight into detrimental side effects of opioids in renal

system. Use of primary human podocytes in this study is a particular strength. The experiments are mostly robust and well-presented, the manuscript is written clearly.

I have a few comments for the authors to consider.

1. Although their conclusion that the Ca²⁺ signals generated downstream of the KOR activation are mediated by TRPC6 is likely to be correct, the evidence is not a 100% watertight. At the concentration used (1-50 uM) SAR7334 would also inhibit TRPC7 and TRPC3 (according to Tocris), functional activity of TRPC3 has been reported in podocytes (Hall, Wang and Spurney, Cells 2019). Thus, I would suggest a more careful language when claiming TRPC6 as KOR's effector.

In relation to the above, why SAR7334 was used at 1 uM in rat experiments (e.g. Fig. 2) and at 50 uM in human cells (Fig. 3)?

Further on this topic, even at 50 uM SAR7334 does not completely block KOR-induced Ca²⁺ transient, hence, there is apparently a TRPC6-independent component in the response. This needs to be acknowledged.

2. DPDPE induced small and slow Ca²⁺ response, at least in some cells (Fig. 1B, C). This needs to be acknowledged.

3. Can you clarify the conditions of human kidneys used for podocyte isolation? These are described in the manuscript as "discarded human kidney transplants", were these kidneys compromised in some way? This is important to clarify - can these podocytes be considered 'healthy'?

4. More caution is needed in interpreting the data from the systemic application of KOR (Fig 5), kidney is not the only organ that controls blood pressure.

5. I would welcome a comment on the possible mechanism of TRPC6 activation by KOR (can be added to Discussion).

Minor.

6. Supplementary video is cool but the cell seemingly just crawled away at the end. Perhaps it didn't like to be watched.

7. Beginning of p. 9: "Importantly, basal [Ca²⁺]_i concentration was elevated in the opioid treated rats on a HS diet (Fig. 5D)." I assume that Ca²⁺ was elevated in podocytes, not rats.

8. P. 11, second para: "The stimulation of κ-ORs with specific agonist shows rapid and transient immobilization of [Ca²⁺]_i in freshly isolated rat glomeruli or human cultured podocytes." Did you mean 'mobilization'? Immobilization doesn't make much sense in this context.

9. Fig. 4C, lower panel - why cell area is expressed in arbitrary units?

Reviewer #2 (Comments to the Authors (Required)):

The present study demonstrated that stimulation of κ -ORs, but not μ -ORs or δ -ORs, evoked $[Ca^{2+}]_i$ transient in podocytes via TRPC6 channels and that IV of κ -OR agonist augmented blood pressure and kidney damages in SS rats. It is concluded that the κ -OR/TRPC6 signaling directly influences podocyte calcium handling, provoking the development of kidney injury in the opioid treated hypertensive cohort.

This interesting study provides a novel insight for the understanding of the role of opioid receptors in podocyte damage and advances our knowledge of the pathogenesis of the development of hypertension and CKD in the context of pain management. The study is well performed and clearly presented. There are some minor comments.

1. Can authors provide rationales for that how the drug concentrations and doses used were determined? What is the pathological relevance of the concentration and/or dose used?
2. Can authors provide information that which or what type of G-protein is involved in the ORs and/or in TRPC6 activation? This information may provide further support to link the OR and TRPC6 action.
3. Is the enhanced damage in kidney due to the augmented hypertension or the direct acting on the renal cells?

Reviewer #3 (Comments to the Authors (Required)):

Comments on "Role of opioid signaling in kidney damage during the development of salt/induced hypertension"

Golosova and coworkers studied the impact of opioid-receptor stimulation in podocytes on the pathogenesis of renal damage in hypertensive conditions. This study is interesting and timely given the crisis of opioid abuse in many countries specially in the context of hypertension and chronic kidney disease. The authors made the following main observations:

- 1) That kappa opioid receptor agonist BRL 52537 promotes an increase in intracellular calcium in podocytes from freshly isolated rats' glomeruli, in human immortalized podocytes and in podocytes from freshly isolated human glomeruli.
- 2) This increase seems to be mediated by Ca^{2+} influx via TRPC6 since the specific TRPC6 inhibitor SAR7334 blunts the BRL 52537-induced Ca^{2+} signal.
- 3) Treatment of SS rats with high salt diet plus BRL 52537 induced an increase in MAP, increased albuminuria and nephrin shedding indicative of podocyte damage corresponding with increased basal intracellular Ca^{2+} in isolated podocytes.

In general, the experiments are well executed and controlled; nevertheless, few issues remain to be resolved.

The conclusion that the observed in vivo phenotype is caused by BRL 52537 effect specifically on podocytes is not supported with experimental data. The evidences provided do not exclude that kappa opioid receptor activation elsewhere causes the observed phenotype. This issue is difficult to solve since it would require podocyte specific deletion of kappa opioid receptor. In any case, a discussion of the limitations of the study and the correlative nature of the findings should be included.

Figure S1 would be central to support the role of opioids in kidney damage. Nevertheless, the graph is likely mislabeled or, in any case, very difficult to understand (e.g y axis on the right is not labeled, a bar with 12% is higher than a bar with 14%?). Moreover, the difference between groups is minor and no statistical significance is reported. Could you please comment on this?

The relative abundance of TRPC6 in rats under high salt diet treated with BRL (Fig 6B) is not significantly different to LS condition. These results suggest that BRL protects podocytes from deleterious increase of TRPC6 leading to DHS/hypertension-induced damage. Likewise, it is also contradictory that basal Ca^{2+} concentration in 14DHS is lower than in 14DHS+BRL (Fig 5D) when the TRPC6 expression in the latter do not differ from LS group (Fig 6B). Increasing the n and including a control group of glomeruli from LS treated rats in experiments from Fig5D would be helpful in this regard. Also, direct experimental evidence (e.g. quantifying the number of podocytes) supporting the argument that a decreased number of podocytes could be the cause of the decreased expression of TRPC6 should be provided.

Table 2: Urinary Na/K ratio in BRL treated rats (LS) is decreased (probably significantly) pointing towards Na retention. This could be an independent cause contributing to the development of kidney damage in BRL treated rats. An assessment of the activity of the RAS (plasma or 24h urinary aldosterone, plasma renin activity or renin mRNA...) in this experiment would be informative.

Figure 3A: Image quality is low and makes it difficult to conclude whether k-OR and TRPC6 are co-expressed in the same cells. An option would be fixed kidney sections of SS rats. This would be an additional evidence suggesting that the observed phenotype could be due to specific effect on podocytes.

The LS labeling is somehow misleading.

The last sentence in the results section is highly hypothetical and should be deleted or down tuned

In Discussion page 11, second paragraph is written that "shows rapid and transient immobilization of Ca^{2+} ". Could it be "mobilization" of Ca^{2+} ?

Statistics in Table 1 is comparing between groups on the same treatment or within the same group with different treatment?

Dear Dr. Bhatt,

Please consider a revised version of our paper entitled “Role of opioid signaling in kidney damage during the development of salt-induced hypertension”, – #LSA-2020-00853-T for publication in *Life Science Alliance*. We would like to thank you and the reviewers for their valued work and very important comments. We understand that it is a very difficult and responsible task and would like to thank everyone for their profound contribution! We ran new experiments and made appropriate revisions to the manuscript to address all of reviewer’s comments. The details are described below. Thank you for considering our work for publication in your journal.

Sincerely yours,
Alexander Staruschenko, PhD

RESPONSE TO REVIEWERS

Reviewer #1

I've read with interest this manuscript from the Alexander Staruschenko's group. The study describes functional expression of kappa opioid receptors in rat and human podocytes and investigates some mechanisms triggered by KOR activation in vitro and in vivo. The authors show that KOR ligand induces Ca²⁺ influx through the TRPC channels in podocytes. A prolonged stimulation results in Ca²⁺ overload and podocyte death which, in turn, compromises the integrity of the glomerular filtration barrier leading to renal damage and loss of blood pressure control. These are significant findings as they provide novel insight into detrimental side effects of opioids in renal system. Use of primary human podocytes in this study is a particular strength. The experiments are mostly robust and well-presented, the manuscript is written clearly.

1. Although their conclusion that the Ca²⁺ signals generated downstream of the KOR activation are mediated by TRPC6 is likely to be correct, the evidence is not a 100% watertight. At the concentration used (1-50 uM) SAR7334 would also inhibit TRPC7 and TRPC3 (according to Tocris), functional activity of TRPC3 has been reported in podocytes (Hall, Wang and Spurney, Cells 2019). Thus, I would suggest a more careful language when claiming TRPC6 as KOR's effector.

Thank you very much for your comment and suggestion. To address your question, we performed additional experiments using 100 nM of SAR 7334, which is specific for TRPC6 (EC₅₀ values are 9.5, 226 and 282 nM for TRPC6, TRPC7 and TRPC3-mediated Ca²⁺ influx), in human immortalized podocytes. 100 nM of SAR 7334 almost completely blocked calcium release in response to BRL 52537 (similar to previously used 1 μM of SAR 7334). Novel data were added to Figure 2E. The role of TRPC channels has been additionally described in the discussion.

In relation to the above, why SAR7334 was used at 1 uM in rat experiments (e.g. Fig. 2) and at 50 uM in human cells (Fig. 3)? Further on this topic, even at 50 uM SAR7334 does not completely block KOR-induced Ca²⁺ transient, hence, there is apparently a TRPC6-independent component in the response. This needs to be acknowledged.

Thank you very much for your question. Unfortunately, the human kidneys are not available now and we are unable to test a different concentration. We agree that SAR 7334 does not completely block KOR-induced calcium transient. Thus, we have acknowledged this in the text.

2. DPDPE induced small and slow Ca²⁺ response, at least in some cells (Fig. 1B, C). This needs to be acknowledged.

According to our data the effect of DPDE is comparatively low. However, we agree that application of DPDPE resulted in small calcium influx, which is now mentioned in the result section.

3. Can you clarify the conditions of human kidneys used for podocyte isolation? These are described in the manuscript as "discarded human kidney transplants", were these kidneys compromised in some way? This is important to clarify - can these podocytes be considered 'healthy'?

Thank you very much for these questions. We used a novel vibrodissociation approach to isolate the human glomeruli, which has been recently described (PMID: 31588797). This approach is gentle and does not use enzymatic activity and results in very little to no damage to the glomeruli. Human kidneys were obtained from an organ procurement company where they were harvested with the intent to be used for transplantation but not used and destined to be discarded. Therefore, such kidneys were considered healthy as they were intended for transplantation. We have a histology of each kidney in this study (data not shown) and these kidneys do not present any evident pathology.

4. More caution is needed in interpreting the data from the systemic application of KOR (Fig 5), kidney is not the only organ that controls blood pressure.

We agree that the kidney is not the only organ that controls blood pressure and there is a possibility that opioids have additional effects in the development of hypertension. This was acknowledged in the discussion. We hope that our data provide strong evidence that using opioids during hypertension promotes podocyte damage and exacerbates chronic kidney disease with blood pressure elevation.

5. I would welcome a comment on the possible mechanism of TRPC6 activation by KOR (can be added to Discussion).

Thank you for this question. We added some speculation about the possible mechanism of activation of TRPC6 in the discussion.

Minor comments.

6. Supplementary video is cool but the cell seemingly just crawled away at the end. Perhaps it didn't like to be watched.

Thank you for this comment. The video shows a cell fragment on the high magnification in 3D. The visualized lamellipodium process was retracted entirely from the field of view after the prolonged exposure with opioid receptor agonist (more than 40 min). This is an example of a strong podocyte foot process cytoskeletal rearrangement.

7. Beginning of p. 9: "Importantly, basal [Ca²⁺]_i concentration was elevated in the opioid treated rats on a HS diet (Fig. 5D)." I assume that Ca²⁺ was elevated in podocytes, not rats.

Thank you very much, this has been corrected.

8. P. 11, second para: "The stimulation of κ -ORs with specific agonist shows rapid and transient immobilization of $[Ca^{2+}]_i$ in freshly isolated rat glomeruli or human cultured podocytes." Did you mean 'mobilization'? Immobilization doesn't make much sense in this context.

Thank you very much, this has been corrected.

9. Fig. 4C, lower panel - why cell area is expressed in arbitrary units?

Thank you very much, this has been corrected.

Reviewer #2 (Comments to the Authors (Required)):

The present study demonstrated that stimulation of κ -ORs, but not μ -ORs or δ -ORs, evoked $[Ca^{2+}]_i$ transient in podocytes via TRPC6 channels and that IV of κ -OR agonist augmented blood pressure and kidney damages in SS rats. It is concluded that the κ -OR/TRPC6 signaling directly influences podocyte calcium handling, provoking the development of kidney injury in the opioid treated hypertensive cohort.

This interesting study provides a novel insight for the understanding of the role of opioid receptors in podocyte damage and advances our knowledge of the pathogenesis of the development of hypertension and CKD in the context of pain management. The study is well performed and clearly presented. There are some minor comments.

1. Can authors provide rationales for that how the drug concentrations and doses used were determined? What is the pathological relevance of the concentration and/or dose used?

Thank you for this comment. The concentration of the drug for our chronic studies has been selected according to the studies performed by Cheng et al. (PMID: 17879026). For glomeruli and immortalized human podocytes, we performed the set of experiments to identify an EC_{50} for BRL 52537 (see figure 2D) for these cells.

2. Can authors provide information that which or what type of G-protein is involved in the ORs and/or in TRPC6 activation? This information may provide further support to link the OR and TRPC6 action.

Thank you very much for this question, we have added the potential explanation of the signaling pathway in the discussion.

3. Is the enhanced damage in kidney due to the augmented hypertension or the direct acting on the renal cells?

Thank you very much for this important question. We hypothesize that opioid use results in elevated calcium levels in podocytes, triggering cell apoptosis and consequent kidney damage, thereby aggravating hypertension. This has been added to the discussion.

Reviewer #3 (Comments to the Authors (Required)):

Comments on "Role of opioid signaling in kidney damage during the development of salt/induced hypertension" Golosova and coworkers studied the impact of opioid-receptor stimulation in podocytes on the pathogenesis of renal damage in hypertensive conditions. This study is interesting

and timely given the crisis of opioid abuse in many countries specially in the context of hypertension and chronic kidney disease. The authors made the following main observations:

1) That kappa opioid receptor agonist BRL 52537 promotes an increase in intracellular calcium in podocytes from freshly isolated rats' glomeruli, in human immortalized podocytes and in podocytes from freshly isolated human glomeruli.

2) This increase seems to be mediated by Ca²⁺ influx via TRPC6 since the specific TRPC6 inhibitor SAR7334 blunts the BRL 52537-induced Ca²⁺ signal.

3) Treatment of SS rats with high salt diet plus BRL 52537 induced an increase in MAP, increased albuminuria and nephrin shedding indicative of podocyte damage corresponding with increased basal intracellular Ca²⁺ in isolated podocytes.

In general, the experiments are well executed and controlled; nevertheless, few issues remain to be resolved.

The conclusion that the observed in vivo phenotype is caused by BRL 52537 effect specifically on podocytes is not supported with experimental data. The evidences provided do not exclude that kappa opioid receptor activation elsewhere causes the observed phenotype. This issue is difficult to solve since it would require podocyte specific deletion of kappa opioid receptor. In any case, a discussion of the limitations of the study and the correlative nature of the findings should be included.

Thank you very much for this important comment. We have added the limitations of the study to the discussion section.

Figure S1 would be central to support the role of opioids in kidney damage. Nevertheless, the graph is likely mislabeled or, in any case, very difficult to understand (e.g y axis on the right is not labeled, a bar with 12% is higher than a bar with 14%?).

Moreover, the difference between groups is minor and no statistical significance is reported. Could you please comment on this?

Thank you very much for your comment. This reviewer is correct, and the graph was mislabeled, which is now adjusted. The explanation of the obtained results has been added to figure legends and in result section. Opioid-induced glomerular injury was elevated but did not reach statistical difference between the two damaged glomeruli populations. This could be due to the rapid glomerular damage increase under the HS diet.

The relative abundance of TRPC6 in rats under high salt diet treated with BRL (Fig 6B) is not significantly different to LS condition. These results suggest that BRL protects podocytes from deleterious increase of TRPC6 leading to DHS/hypertension-induced damage. Likewise, it is also contradictory that basal Ca²⁺ concentration in 14DHS is lower than in 14DHS+BRL (Fig 5D) when the TRPC6 expression in the latter do not differ from LS group (Fig 6B). Increasing the n and including a control group of glomeruli from LS treated rats in experiments from Fig5D would be helpful in this regard. Also, direct experimental evidence (e.g. quantifying the number of podocytes) supporting the argument that a decreased number of podocytes could be the cause of the decreased expression of TRPC6 should be provided.

Opioid treatment rapidly increases podocyte damage according to our data. This results in the loss of structural integrity of podocyte foot processes and corresponding decrease in cell surface area

and cell numbers. The observed nephrin shedding clearly illustrates this fact. Relative expression of TRPC6 (per area or per cell) in this case will be significantly higher. Unfortunately, we do not have samples to perform analysis in a NS group or to quantify the number of podocytes. However, we have discussed this in the result and discussion sections.

Table 2: Urinary Na/K ratio in BRL treated rats (LS) is decreased (probably significantly) pointing towards Na retention. This could be an independent cause contributing to the development of kidney damage in BRL treated rats. An assessment of the activity of the RAS (plasma or 24h urinary aldosterone, plasma renin activity or renin mRNA...) in this experiment would be informative.

Thank you very much for you comment and suggestion. We apologize for a confusion. We have revised this and updated our protocol and the Tables labels. BRL was administered on the first day of HS challenge. The difference in urinary and plasma electrolytes is not significantly different on NS since it represents the group of animals not treated with opioids. An assessment of the RAAS activity is critical, but we believe it is outside of scope of this manuscript. We would like to acknowledge this in our future project, which is currently in progress.

Figure 3A: Image quality is low and makes it difficult to conclude whether κ -OR and TRPC6 are co-expressed in the same cells. An option would be fixed kidney sections of SS rats. This would be an additional evidence suggesting that the observed phenotype could be due to specific effect on podocytes.

There are multiple reports from our laboratory and others demonstrating the critical role of TRPC6 and its expression in the podocytes. The discussion of TRPC6 expression has been added (PMID: 31416818, PMID: 20685822) in the result section. In addition, we have performed an immunohistochemical staining of κ -OR protein in human kidney (new Figure 3B) showing its expression in the podocytes.

The LS labeling is somehow misleading.

LS has been changes to normal salt (NS) in the text, tables and figures.

The last sentence in the results section is highly hypothetical and should be deleted or down tuned

Thank you very much, this has been corrected.

In Discussion page 11, second paragraph is written that "shows rapid and transient immobilization of Ca^{2+} ". Could it be "mobilization" of Ca^{2+} ?

Thank you very much, this has been corrected.

Statistics in Table 1 is comparing between groups on the same treatment or within the same group with different treatment?

Thank you very much for this question. The statistics in the table is the comparison between groups (Group 1 vs. Group 2, vehicle vs. BRL-treated) within the same NaCl diet. There was no BRL treatment on NS. We changed the labels of the groups and provided additional explanation.

September 28, 2020

RE: Life Science Alliance Manuscript #LSA-2020-00853-TR

Dr. Alexander Staruschenko
Medical College of Wisconsin
Physiology
8701 Watertown Plank Road
Milwaukee, WI 53226

Dear Dr. Staruschenko,

Thank you for submitting your revised manuscript entitled "Role of opioid signaling in kidney damage during the development of salt-induced hypertension". We would be happy to publish your paper in Life Science Alliance pending final revisions requested by the LSA editors (below) and the reviewer(s) (appended at the end of this email), and edits necessary to meet our formatting guidelines.

- Please provide source data for Figure 2A
- Please add cut out boxes for the middle and right top images in Figure 3A
- Please provide scale bars for Fig S1A
- Please provide a pbp response to the concerns raised by reviewer 3 (appended at the end of this email)

A. FINAL FILES:

-- Summary blurb (enter in submission system): A short text summarizing in a single sentence the study (max. 200 characters including spaces). This text is used in conjunction with the titles of

papers, hence should be informative and complementary to the title. It should describe the context and significance of the findings for a general readership; it should be written in the present tense and refer to the work in the third person. Author names should not be mentioned.

B. MANUSCRIPT ORGANIZATION AND FORMATTING:

Sincerely,

Shachi Bhatt, Ph.D.
Executive Editor
Life Science Alliance

Reviewer #3 (Comments to the Authors (Required)):

Comments on "Role of opioid signaling in kidney damage during the development of salt/induced hypertension". Golosova and coworkers have satisfactorily addressed most of the concerns raised in the first revision.

In the opinion of this reviewer, the manuscript improved a lot and only few minor issues remain to be solved.

- Reference missing in the following sentence in introduction (Page 3): "The extensive use of opioid-based pain management strongly correlates with poor cardiovascular and cardiorenal outcomes, including increased albuminuria and reduced glomerular filtration rate; as such, opioid medication is strongly linked to progressive renal damage and blood pressure elevation"
- IF quality in figure 3A is still unsatisfactory. I was not convinced neither by the explanation of the authors, nor by the newly included Figure 3B since the staining does not allow the identification of the cells expressing k-OR. This issue does not invalidate the findings since it was clearly proved that TRPC6 inhibition blunts the increase in Ca²⁺ triggered by k-OR agonist. But this figure does not match the quality of other experiments in the manuscript and the experimental reputation of the authors.
- Please indicate water and food consumption of animals under high Na diet. In the same line, differences in plasma and urinary electrolytes should be briefly discussed since this would not be expected just due to glomerular damage.
- Figure 7 suggests that opioids promote the transition from normotension to hypertension although this was not addressed in the manuscript. The left panel would be sufficient to summarize the findings.
- The following sentence in Discussion (Page 10) should be revised for meaning: "Podocytes play a key role in the maintenance of glomerular structure, control permeability, and when damage affects progression of glomerulopathies"

Dear Dr. Bhatt,

Please consider a revised version of our paper entitled "Role of opioid signaling in kidney damage during the development of salt-induced hypertension", – #LSA-2020-00853-T for publication in *Life Science Alliance*. We made appropriate revisions to the manuscript to address the reviewer and editor's comments. The details are described below.

Sincerely yours,

Alexander Staruschenko, PhD

RESPONSE TO REVIEWERS

Reviewer #3

Reference missing in the following sentence in introduction (Page 3): "The extensive use of opioid-based pain management strongly correlates with poor cardiovascular and cardiorenal outcomes, including increased albuminuria and reduced glomerular filtration rate; as such, opioid medication is strongly linked to progressive renal damage and blood pressure elevation"

Thank you very much, this has been corrected.

IF quality in figure 3A is still unsatisfactory. I was not convinced neither by the explanation of the authors, nor by the newly included Figure 3B since the staining does not allow the identification of the cells expressing k-OR. This issue does not invalidate the findings since it was clearly proved that TRPC6 inhibition blunts the increase in Ca²⁺ triggered by k-OR agonist. But this figure does not match the quality of other experiments in the manuscript and the experimental reputation of the authors.

Thank you very much. Figures 3A and B have been moved from the main text to supplementary material (new Fig. S1). Cut out boxes for the middle and right top images also added as requested by the editor.

Please indicate water and food consumption of animals under a high Na diet. In the same line, differences in plasma and urinary electrolytes should be briefly discussed since this would not be expected just due to glomerular damage.

Thank you for your comment. Rats were provided high salt diets, and water ad libitum as indicated in the methods section. Unfortunately, water and food consumption was not evaluated. Changes in plasma and urinary electrolytes have been acknowledged in the results section.

Figure 7 suggests that opioids promote the transition from normotension to hypertension although this was not addressed in the manuscript. The left panel would be sufficient to summarize the findings.

Thank you for your comment. We updated Figure 7 as requested.

The following sentence in Discussion (Page 10) should be revised for meaning: "Podocytes play a key role in the maintenance of glomerular structure, control permeability, and when damage affects progression of glomerulopathies"

Thank you very much, this has been revised.

October 2, 2020

RE: Life Science Alliance Manuscript #LSA-2020-00853-TRR

Dr. Alexander Staruschenko
Medical College of Wisconsin
Physiology
8701 Watertown Plank Road
Milwaukee, WI 53226

Dear Dr. Staruschenko,

Thank you for submitting your Research Article entitled "Role of opioid signaling in kidney damage during the development of salt-induced hypertension". It is a pleasure to let you know that your manuscript is now accepted for publication in Life Science Alliance. Congratulations on this interesting work.

DISTRIBUTION OF MATERIALS:

Again, congratulations on a very nice paper. I hope you found the review process to be constructive and are pleased with how the manuscript was handled editorially. We look forward to future exciting submissions from your lab.

Sincerely,

Shachi Bhatt, Ph.D.

Executive Editor

Life Science Alliance

<https://www.life-science-alliance.org/>
